# Geography of Indian Butterflies: Patterns Revealed by Checklists of Federal States

**DOI:** 10.3390/insects14060549

**Published:** 2023-06-13

**Authors:** Gaurab Nandi Das, Zdenek Faltynek Fric, Shristee Panthee, Jatishwor Singh Irungbam, Martin Konvicka

**Affiliations:** 1Faculty of Sciences, University of South Bohemia, 370 05 České Budějovice, Czech Republic; gaurab68nandidas@gmail.com; 2Biology Centre CAS, Institute of Entomology, 370 05 České Budějovice, Czech Republic; fric@entu.cas.cz; 3CAS Key Laboratory of Tropical Forest Ecology, Xishuangbanna Tropical Botanical Garden, Chinese Academy of Sciences, Mengla 666303, China; shristee@xtbg.ac.cn; 4University of Chinese Academy of Sciences, Beijing 101408, China; 5Sphingidae Museum, Orlov, 261 01 Pribram, Czech Republic; jatishwor.irungbam@gmail.com

**Keywords:** biogeographic elements, climate, faunal turnover, latitudinal gradient, Oriental realm, peninsular effect

## Abstract

**Simple Summary:**

The Republic of India is a tropical megadiverse country, encompassing four global biodiversity hot-spots and harboring 1379 butterfly species within its borders. Despite a long history of documenting Indian biodiversity, there are extremely few quantitative studies that have analyzed species-richness patterns within the country. We capitalized on the division of the country into 36 federal states and territories, and compiled and revised per-states butterfly checklists. Then, we analyzed the species-richness patterns, as well as the richness of endemic species and the numbers of species belonging to main biogeographic elements with respect to geography, climate, land covers and socioeconomic conditions of the administrative units. Such common macroecological predictors as area, latitude and land covers diversity did not affect the species richness, whereas the topographic diversity and energy availability had major effects. This is due to the peculiar biogeography of India, i.e., the peninsula narrowing towards the Equator, being isolated from the northern mainland by high mountains, and connected to species-rich eastern regions by only a narrow conduit. In multiple regression models, geographic variables were the strongest predictors of species-richness patterns, followed by climate and land covers. Our approach can be used as an initial step toward understanding distribution patterns in those regions of the world, for which detailed distribution data are not yet available but checklists per administrative units exist.

**Abstract:**

Butterflies are widely used to analyze biogeographical patterns, both at the global and regional scales. Thus far, most of the latter originated from well-surveyed northern regions, while the species-rich tropical areas lag due to a lack of appropriate data. We used checklists of 1379 butterfly species recorded in 36 federal states of the Republic of India (1) to explore the basic macroecological rules, and (2) to relate species richness and the distribution of endemics and geographic elements to geography, climate, land covers and socioeconomic conditions of the states. The area, land covers diversity and latitude did not affect species richness, whereas topographic diversity and the precipitation/temperature ratio (energy availability) were positive predictors. This is due the geographic and climatic idiosyncrasies of the Indian subcontinent, with its highest species richness in the small, densely forested mountainous northeast that receives summer monsoons. The peninsular effect that decreases the richness towards the tip of subcontinent is counterbalanced by the mountainous forested Western Ghats. Afrotropical elements are associated with savannahs, while Palearctic elements are associated with treeless habitats. The bulk of Indian butterfly richness, and the highest conservation priorities, overlap with global biodiversity hotspots, but the mountainous states of the Western Himalayas and the savannah states of peninsular India host distinctive faunas.

## 1. Introduction

In an ideal world, large-scale analyses of species distribution patterns would be based on data obtained using a standard survey of equally-sized spatial units, typically grids; this allows for direct comparisons among taxa, countries, continents and biogeographical realms [1,2,3]. For obvious reasons, data of such quality are available only for a few superbly studied taxa [4,5,6,7], usually for selections from much-studied countries and regions, typically situated in the Global North [8,9,10,11,12]. Methods to overcome these limitations include the use of point records transferred to grid systems and modeling species distributions for non-surveyed locations [8,11,13], ideally accounting for variations in survey efforts [14]. An alternative is not to work with regular-sized grids, but with spatially well-defined though irregularly-sized and shaped units, such as states or administrative provinces, for which species distribution data are traditionally recorded [15,16,17,18,19]. Relating species distribution patterns to political borders has the direct advantage that conservation policies or environmental education can be typically organized within such units [18,20].

The Republic of India is a large tropical country; it is one of the 17 megadiverse countries in the world, harboring four global biodiversity hotspots within its borders [21]. Already at the beginning of the 20th century, Blanford [22] tried to summarize the biogeography of Indian fauna, demonstrating that Oriental elements dominate the lower altitudes of the whole landmass due to the extensive interchange with more easterly areas via northeastern India. The Garo-Rajmahal gap in present Bangladesh was much-discussed as a conduit, as well as a barrier for faunal interchange [23,24,25]. The Palearctic faunal elements are largely restricted to high altitudes of the Himalayas, plus partially to western deserts [24]. Regarding the Afrotropical elements, some interchange happens via the southern edge of the Eurasian landmass, oversea dispersal, or temporary land bridges [26]. Importantly, the southwestern volcanic continental margins (i.e., the Western Ghats) of the Indian peninsula play an important role as a secondary speciation center with a high endemism rate [24]. The Western Ghats also create a rain shadow zone in the Deccan Plateau by influencing the monsoon climate [27].

The diversity of its relief and biogeographic features [21] makes India one of the most butterfly-rich countries in the world [28]. Building on a long taxonomy and faunistic tradition [29,30], the first biogeography synthesis appeared in the 1960s, when Holloway [31,32] adopted cluster analysis to explore the geographical affinities of selected Indian taxa. He expectably found a prevalence of the Austro-Oriental faunal element, followed by Afrotropical and Palearctic elements; endemism was highest in the mountainous northeast of the country, whereas peninsular India was relatively species-poor. Later, Kunte [33] summarized the biogeographic affinities of Indian butterflies inhabiting the Western Ghats hotspot (332 species, ≈22% of Indian fauna). Recently, there has been an explosion of diversity patterns studies conducted at smaller scales, with a notable bias towards biodiversity hotspots (e.g., Western Ghats [34], Eastern Himalaya [35,36]). Species-poorer peninsular India remains neglected [37,38]. Reflecting the explosive increase in knowledge, new comprehensive checklists on Indian butterflies were produced by Varshney and Smetacek [39] and Gasse [40]. Unlike earlier efforts, these books summarized the distributions of all 1379 Indian species by federal states, allowing for the first time the analysis of distribution patterns on this scale.

The general macroscale patterns, repeatedly disclosed for terrestrial animals including butterflies, show a poleward decrease in species richness [18,41,42,43], an increase in richness with habitat/relief diversity [44,45,46,47] and with available energy [48,49,50], and a relatively high proportion of endemics in mountains and on islands [51,52,53]. Furthermore, with particular relevance to the Republic of India, the numbers of species tend to decrease towards the tips of peninsulas [54,55,56]. Most of this information was derived from studies conducted in northern regions despite the bulk of global species richness; hence, high conservation responsibilities were distributed throughout tropical and subtropical areas [18,57].

In the present study, we examined the diversity pattern of butterflies across Indian federal states using the state-by-state presence data. With these data, we first tested for validity of the selected basic macroecology patterns. Secondly, we used regression analysis to explore factors that were putatively responsible for species richness, richness of the main biogeographic elements, distribution of endemic species and species turnover, the latter being a measure of uniqueness of individual states’ faunas. We constructed separate models for four sets of predictors, describing physical geography, climate, land cover and the socioeconomic conditions of the states; we compared their relative explanatory power and explored their mutual relations. Such a study has never been performed for an entire Insect taxon in India, as were carried out for mammals (Karanth [58]) and birds (eBird [59]). In mammals, the lowest diversity was found to be in northern mountainous Ladakh, whereas it was highest in the peninsula and especially along India’s western coast; moreover, the highest diversity of primates was found to be in the eastern rainforest states, similarly to birds. We believe that our research will provide a baseline for future comparisons with other groups of organisms, but also with other tropical regions, and for conservation planning and assessments of future faunal changes.

## 2. Materials and Methods

### 2.1. Data Sources and Digitization

We compiled a matrix of all Indian butterfly species and their presences in all Indian federal states, including union territories (hereafter states; *n* = 36: https://knowindia.india.gov.in/states-uts/ (accessed on 8 May 2023)) (Figure 1, Appendix A). To compile the butterfly records, we used the recent checklists by Varshney and Smetacek [39] and Gasse [40], which were updated with more recent records. Note that we included a few species that needed clarification of their current status in India, such as *Pontia sherpae* (Epstein, 1979), or *Ypthima sobrina* Elwes and Edwards, 1893. The higher classification followed that of van Nieukerken et al. [60]; the taxonomy on family-, genus-, and species-levels followed Eliot [61], Ackery [62], Lang [63], Wahlberg et al. [64,65], Toussaint et al. [66], Savela [67], Bálint [68] and Inayoshi [69]. The series “Guide to the Butterflies of the Palearctic Region” edited by G.C. Bozano and “Butterflies of the Palearctic Asia” edited by V. Tshikolovets were also consulted.

Next, we attributed each species to a biogeographic element, distinguishing Afrotropical (including Madagascar), Oriental (including Sino-Japanese, Australian and Papuan region), Palearctic (including both East and West Palearctic) and Cosmopolitan species. We also distinguished species that were endemic to the Republic of India, defined as those with a distribution restricted to the Indian territory.

We calculated the mean turnover as a measure of dissimilarity in faunal composition among units, i.e., Indian states in our case. We calculated the values as a mean value of state-by-state species composition changes, using the “nestedbetajac” function of the R “vegan” package [70].

For predictors describing the states (Table 1, Appendix A), we distinguished (1) geography, which includes three cardinal coordinates of the states’ centroids, area and altitude difference between the lowest and highest points; (2) climate, used from CHELSA v.1.2 [71] with a spatial resolution 2.5 min; the bioclimatic variables were extracted for the area within the boundaries, and from these, we used average values; (3) land covers, taken from official publications by the Forest Survey of India (hereafter FSI [72]), reporting covers of main habitats per individual states; and (4) socioeconomic predictors, which included population data, including per capita GDP, literacy rate and data on livestock ownership reported by the FSI [72] and Reserve Bank of India [73]. Figure 1c presents the interrelationships among the predictors, which were visualized using principal component analysis (PCA) in CANOCO v. 5.15 [74]. Appendix A presents a matrix of Pearson’s correlations among the predictors.

### 2.2. Statistical Analyses

Our dependent variables were the per-state richnesses of all the Oriental, Afrotropical and Palearctic species, the numbers of endemic species, and species turnover. We did not consider the numbers of cosmopolitan species due to the low variations among the states (coefficient of variation: =0.15).

Prior to the analyses, we standardized both the dependent variables and predictors to zero mean and unit variance, in order to achieve comparability of the regression coefficients. We used generalized linear model (function *glm*) regressions with a Gaussian link, selecting from candidate models using the information theory approach (Akaike information criterion). We constructed separate models for geography, climate, land covers and socioeconomic variables. Except for climate, we proceeded as follows.

To explore basic macroecological patterns, we computed generalized linear models relating richness of all, endemic, oriental, Afrotropical and Palearctic species to area, precipitation/temperature ratio, altitude difference and latitude of the states.

In order to construct more complex models for the geography, climate, land covers and socioeconomic predictors, we started with separate regressions for of the potential predictors, plus their second-order polynomials and first-order interactions with all of the other predictors. From these, we selected those predictors and their first-order interactions that separately decreased the AIC relative to a null model y ~+1 by >≈2.0, and defined a saturated model containing all of those model terms. Then, we simplified this saturated model using stepwise backward elimination (drop1 function), until all of the predictors retained in the model diminished the AIC value compared to a higher-order model and significantly differed from one another. In the cases of several competing models with very similar AICs, we selected the model with the lowest number of terms (i.e., the highest residual degrees of freedom).

For climate, we used principal component analysis (PCA), computed again in CANOCO v. 5.15 [74] to reduce the high number (*n* = 19) of bioclimatic variables into four principal axes. These, used as novel composite predictors, ran from states with hightemperature seasonality and low precipitation towards those with low seasonality and high precipitation (Clim1, eigenvalue 0.870); from states with high diurnal and seasonal temperature differences towards those with low diurnal differences (Clim2, 0.059); from states with a diurnally or seasonally variable climate towards those with a stable climate (Clim3, 0.036); and from states with high values of precipitation, even in their driest periods, towards those with low precipitation (Clim4, 0.024) (details: Appendix A).

All the analyses, except for the PCA, were prepared in R software version 4.3.0 [75].

## 3. Results

The species x states matrix contained a total of 1379 butterfly species, 74 of which were endemic to the Republic of India (Figure 1). Of these, 1143 were Oriental elements (82.9%), 206 (14.9%) were Palearctic elements and 23 (1.7%) were Afrotropical elements. More than two-thirds of the species were recorded from northeastern states (or northeastern India, hereafter NE India), followed by Himalayan states. This was reflected in the ranking the most speciose states, which were Arunachal Pradesh (745, or 54% of all Indian species), followed by West Bengal, Manipur and Sikkim (≈53% each), and Meghalaya (52%). The lowest species richness was found in small-sized union territories or islands, i.e., Lakshadweep (1%); Daman, Diu, Dadra and Nagar Haveli (3%); and Puducherry (4%) (Figure 1).

### 3.1. Exploring Basic Biogeographic Patterns

Area alone had no effect on species richness, and the same applied for the endemic, Oriental and Palearctic elements; in contrast, the numbers of Afrotropical elements increased with state area (Table 2). From the two proxies for heterogeneity, altitude difference increased the per-state species richness, and the richness of Oriental and Palearctic elements. Land cover diversity, in contrast, decreased the per-state species richness and the number of Oriental species, and tended to decrease the endemic and Afrotropical species numbers. Latitude did not have any effect at all on Oriental and Afrotropical species. This lack of a relationship effectively refuted a poleward species richness decrease, which manifested as a negative relationship. It also refuted a peninsular effect, which manifested as a positive relationship. However, endemic species increased towards the south, and Palearctic species increased towards the north. The precipitation/temperature ratio, as a measure of available energy, increased the species richness of all groups except for the Palearctic and endemic species. It was the strongest predictor for total species richness, and the richness of Oriental species.

### 3.2. Models for Species Richness

For species richness, the selected geographic model explained >70% of variations in the data. It was followed by the climatic model (54%), whereas the land covers model was rather weak (20%). Socioeconomic predictors explained a higher proportion of variation than the land covers predictors (40%).

The geographic model pointed to an increase in species richness towards the east and higher altitudes, a decrease towards the south, and an interaction latitude x altitude, standing for increased species numbers in southern mountainous states, and decreased numbers in northern lowland states (Figure 2, Table 3). The climatic model contained a single predictor, Clim2, which pointed to a high butterfly richness in states with high precipitation during warm times of the year, i.e., NE India with prominent summer monsoon. The land covers model revealed an increase in richness in states with a high proportion of very dense forests, i.e., the northeastern Arunachal Pradesh, but also NW Himalayan states (Uttarakhand, Jammu and Kashmir) and southernmost Western Ghat states (Kerala, Tamil Nadu), and a decrease in states with high proportions of non-forest (Rajasthan). Finally, the socioeconomic model pointed to increased species richness in states with a high proportion of rural population (NE India, but also eastern India, e.g., Orissa and Bihar, and NW Himalayan states, such as Himachal Pradesh) and a decreased species richness in states with high livestock populations (the Deccan region, most prominently Madhya Pradesh).

### 3.3. Endemic Species

Modeling the per-state numbers of species endemic to the Republic of India returned a complex geographic model, explaining a slightly lower proportion of variation than the model for species richness (65%). It contained a hump-shaped response to longitude, a U-shaped response to latitude, an increase with mean altitude, and a complex latitude x altitude interaction (Table 3). All of this was due to the high endemism in southern mountainous states, i.e., the states of Western Ghats (Kerala, Tamil Nadu and Karnataka), and a few endemics of NE India. The climatic model explained half of the variation relative to the model for entire species richness (≈20%). The numbers of endemics increased with the Clim1 composite variable, i.e., towards states with low seasonality and high precipitation, which are again the Western Ghats states. The land covers model, which was the weakest in terms of explained variation (≈17%), predicted few endemics in states with high representations of non-forest, and many endemics in states with a high cover of scattered trees, again pointing to the western peninsular/Western Ghats states (e.g., Maharashtra, Karnataka). The socioeconomic model explained half of variation relative to the species richness model (37%), pointing to more endemics in states with a high GDP per capita, but a low human population density and low livestock numbers, i.e., the Western Ghats region.

### 3.4. Oriental Elements

The models for geography and land covers contained similar predictors and reached comparable levels of explained variation as the models for all species, which was expected given that Oriental elements represented 82.9% of Indian fauna (Table 4). The geographic model (70% of variation) predicted higher richness towards the east and higher altitudes, i.e., in the NE states. The climatic model (63%) contained predictors Clim1, Clim2 and Clim4, and revealed increases in species numbers towards states with high precipitation, even in the driest months of the year and with low seasonality, i.e., NE states. The land covers model (22%) revealed an increase in states with high proportions of very dense forest (NE states, but also the Western Ghats region and Himalayan states), and decreases in states with non-forest (Rajasthan, Uttara Pradesh and Gujarat). Finally, the socioeconomic model (≈40%) pointed to increases in states with a high rural population (NE India) and decreases in states with high livestock populations and high literacy rates (Rajasthan, Uttar Pradesh).

### 3.5. Afrotropical Elements

The geographic model was complex, containing a quadratic decrease with longitude, an increase with altitude, and a U-shaped response to latitude. Thus, this model accounted for a higher representation of Afrotropical elements in western (Karnataka, Maharashtra) and northwestern (Rajasthan, Madhya Pradesh) states. The climatic model contained negative response to Clim1 and Clim2, i.e., low numbers of Afrotropical elements in states with high precipitation and low seasonality. The land covers model revealed an increase in states with a high proportion of scattered tree patches, i.e., the states of west–central peninsular India (i.e., Maharashtra, Rajasthan and Madhya Pradesh; see Table 4). The socioeconomic model pointed to increases in states with high road densities, high livestock populations and high GDPs per capita, which were again the states in the western–central peninsula.

### 3.6. Palearctic Elements

The geographic model explained an extremely high proportion of variation (97%), pointing to increases in these northern elements, with the highest representation in Himalayan states, with latitude and altitude, and a decrease towards the east (Table 4). The climatic model (42%) revealed increases in Palearctic elements in regions with high seasonality and low precipitation (Clim1), and an increase with climatic stability (Clim3), pointing mainly to the Himalayan state of Ladakh. The land covers model was rather weak (29%), pointing to increased representations of Palearctic elements with moderately dense forests and non-forests, i.e., towards the northern mountains. The socioeconomic model (40%) pointed to increases in states with a high proportion of rural population, high GDP per capita, and low livestock numbers and literacy rates, which again point to the NW Himalayan region.

### 3.7. Species Turnover

The raw values for species turnover, which describes the uniqueness of state faunas compared to other states, pinpointed the NW Himalayan region as being exceptional within India (Figure 3). The selected geographic model (80%) was complex, however, as it included polynomial and interactive relationships between altitude and latitude caused by faunal idiosyncrasies or the Western Ghats and NE India (Table 3). The climatic model, which was much weaker (37%) than the geographic model, revealed distinctive faunas in states with a stable climate (Clim3) and higher precipitation rate (Clim2) (NE India, Western Ghats and Himalayan regions). The still weaker (≈23%) land covers model pointed to a distinctive position of states with a high proportion of non-forest and low proportion of scattered tress, which are again the NW and Himalayan states. The slightly stronger socioeconomic model (≈30%) pointed to peculiar fauna in states with low livestock numbers and low literacy rates (Ladakh in NW, Arunachal Pradesh in NE).

## 4. Discussion

The biogeographic patterns presented here are based on the entire fauna of 1379 Indian butterflies and cover the entire Republic of India, whereas earlier studies worked either with a limited selection of species [32] or were restricted to smaller regions [33]. On the other hand, the earlier authors (e.g., Holloway [32], Wynter-Blyth [76], Mani [77]) included the independent nations of Pakistan, Myanmar, Bangladesh, Nepal, Bhutan and Sri Lanka in their studies, which was not possible in our case due to a lack of similarly detailed data for these countries. In agreement with earlier authors, the species-richest part of India found was the eastern Himalayan region and NE India, which also included the species-richest state, Arunachal Pradesh (745 species), owing to its increased number of Oriental elements. It was followed by the Western Himalayan and Western Ghats regions. Notably, the species richness of several states, namely Mizoram, Tripura (NE India) and Telangana (central/Deccan India), were lower by 20–30% compared to their immediate neighbors, suggesting that these states are under-explored. The former two are geographically remote and sparsely populated, whereas the third was established by splitting from Andhra Pradesh only in 2014; hence, some records from its territory were likely ascribed to Andhra Pradesh in earlier publications.

The models based on geographic predictors were the strongest in terms of their explained variation for all species, endemic species, biogeographic elements and even species turnover. Except for the Afrotropical elements, the species numbers were entirely unrelated to the state areas, indicating that other geographic features were much more important. Additionally, the entire species’ richness did not monotonously decrease with latitude, which had no significant separate effect (Table 2); this refuted the poleward decrease in species richness observed for multiple taxa across the globe [41]. Instead, a prominent longitudinal increase in species numbers towards NE India exists. The decreasing richness towards the southern tips of peninsulas is a well-established phenomenon for butterflies [32] and other terrestrial taxa [78,79,80,81]. This peninsular effect is attributable to decreasing diversity with increasing distance from speciation and/or colonization sources [82,83,84].

The biogeographic idiosyncrasy of the Republic of India, with its bulk of butterfly species belonging to the Oriental element, but only a narrow connection to the rest of the Oriental realm through the Garo-Rajmahal gap in Bangladesh (cf. Hora [23]), explains the decrease in species numbers in the southwestern direction; this is reflected in the importance of the longitudinal gradient in the geography-based models. In the north, the Himalayas represent a formidable barrier that restricts the northward expansion of tropical species; it is also a strong environmental gradient, restricting cold-adapted Palearctic species to mountainous states such as Ladakh (cf. Tshikolovets [85]). A possible alternative dispersal route for Palearctic elements, via the Iranian plateau, is complicated by the arid Thar desert, which is hospitable for just a handful of species that are adapted to extremes of aridity [86,87,88].

The Afrotropical elements displayed a decrease in species richness towards the East, revealing a connection with westerly situated African and Arabian regions that are well known, e.g., for genera *Belenois* Hübner, 1819, *Eurema* Hübner, 1819 and *Tarucus* Moore, 1881 (cf. Basu et al. [89], Irungbam et al. [90]), and with lower numbers in mountainous states (i.e., lower numbers in southwestern Kerala than in south–central Telangana). Their positive responses to states’ areas were due to the positions of the largest federal states (Rajasthan, Maharashtra) in the west of the country.

For all, Oriental and Palearctic elements, the species richness strongly increased with an altitude difference in the single-term models, pointing to positive richness: heterogeneity relationship. Altitude also impacted the geographic models for all of the dependent variables considered. Its interaction with latitude for all and oriental species represented increased species richness in the Western Ghats, the mountainous rainforest region in southern India, and stood out from the less diverse areas that bordered it [24,33]. The Western Ghats are a well-recognized Paleogene forest refugium [91] and speciation center (mammals: Moore [92], Nameer [93]; birds: Ramesh et al. [94]; reptiles: Varadaraju [95]; amphibians: Dutta et al. [96]; Odonata: Subramanian et al. [97]), harboring multiple butterfly endemics [33,98]. By increasing their diversity near the southern tip of the peninsula, they counteract the peninsular decrease in species numbers; however, this effect is not strong enough to generate increased species richness towards the south.

Recall that we considered as endemics only those species that were restricted to the Republic of India, and not species with small ranges, but transcending state borders. This contributed to the prominent role of the Western Ghats states in the models for endemic species (respective proportions of endemics: Kerala and Tamil Nadu 0.10, Karnataka 0.09). These were followed by the Andaman Islands with a 0.05 proportion of endemics. Especially in the NE states, there were numerous multiple species with very small “endemic” ranges extending to neighboring countries, e.g., *Cyllogenes janetae* de Nicéville, 1887 (extending to Bhutan and China: Tibet), *Euaspa motokii* Koiwaya, 2002 or *E*. *mikamii* Koiwaya, 2002 (both extending to N Myanmar) [99,100]. The inclusions of such species among endemics, or extending our analyses to neighboring states, highlight the endemism of the NE states [101].

All of these observations illustrate a complex biogeographic history of Indian butterfly fauna, which attained elements from more easterly parts of the Oriental realm, as well as from the neighboring Palearctic and more distant Afrotropical regions at various times since the Pliocene–Pleistocene [102]. The Himalayas served as a route for northern faunal movement, whereas NE India (Assam and other northeastern states) is the gateway to southeast Asia [24]. The opposite scenarios are also plausible, with the colonization of eastern regions by species originally from India (the “out of India hypothesis”, cf. Karanth [103], Datta-Roy and Karanth [102]), although the evidence from multiple animal groups suggests that this is less likely [104]. Phylogeographic analyses of selected taxa should explore the two hypotheses for butterflies.

The Per-state species richness of both all and Oriental elements strongly responded to climatic factors, and perhaps more importantly, to our extremely simplified measure of energy availability. Thus, while the latitudinal richness pattern does not apply for India, the energy availability hypothesis holds there perfectly, in agreement with authors who view the former as just an outer expression of the latter (e.g., Bonn et al. [49]). In the special case of India, the peninsular effect hypothesis complicates the matter, and more sophisticated modeling studies are needed to decipher the three competing explanations. The states with the highest energy availability are those with high precipitation during the warmest parts of year, i.e., with strong summer monsoons. The summer monsoon region extends beyond the Republic of India, towards Southeast Asia, and the ranges of many NE India butterflies extend in this direction [32]. Complementarily, the drier conditions in peninsular states likely restrict the distribution of many species to NE federal states. The climatic patterns for Afrotropical elements were the opposite of the Oriental ones, with numbers of such species increasing towards climates with large seasonal and diurnal differences. These states, which are situated in SW India, are characterized by a decoupling of the hottest and wettest months, with the highest precipitation later in the year than the highest temperatures. The Palearctic species were expectably quite distinct from other elements. They were restricted to higher latitudes and climates with temperature seasonality and generally low precipitation.

The land cover models were always weaker than the geographic models, except for Afrotropical elements. Contrary to many studies conducted on small spatial scales (e.g., Jeanneret et al. [105], Mukherjee and Mondal [106], Slancarova et al. [107]), land cover diversity decreased rather than increased the numbers of all and Oriental elements. This contradictory effect can probably be explained by the different perceptions of “habitat diversity” at different scales. In studies at the landscape scale, targeting such units as farmlands, nature reserves or protected areas, land cover categories may represent edges, woodland openings, stream banks or similar structures, the densities of which directly transfer to the availability of resources for individual insect species [108]. The land covers recognized by the FSI [72] are mapped on a much broader scale, corresponding to entire biomes. Then, if a state contains a single biome that is rich in species, it may host more butterflies than a state containing several biomes which are species-poorer. The latter point was evident in the increase in all and Oriental elements with very dense forests, i.e., in NE India, the Western Ghats and Himalayan states. The former two are states containing rain forest, a biome that globally hosts more butterflies than other habitats [109]. For Afrotropical elements, species richness increased with scattered trees, i.e., with seasonal savannahs [110] or farmlands/rangelands with abundant trees and shrubs (Appendix A). 

Such species are, for instance, *Belenois aurota* (Fabricius, 1793), distributed from Southern Africa through Arabian peninsula to India, or several species of *Azanus* Moore, (1881) and *Taurucus* Moore, 1881. The rather complex model for Palearctic species pointed to their high numbers in non-forested habitats, which is likely linked to mountainous and grassland characters or adjacent Palearctic locations (Iranian and Tibetan plateau). Compared to tropical realms, a disproportionate number of Palearctic butterflies are grassland dwellers (cf. Tshikolovets [111]), which was reflected in the habitat requirements composition of the subsample of those that reached India.

The socioeconomic predictors of species diversity were necessarily correlational, not pointing to causative factors. Still, they related the distribution of biodiversity to that of human activities, and may allow for educated guesses regarding future pressures on their natural habitats. Among Indian federal states, NE India, with the highest numbers of all and Oriental butterflies, had the highest proportions of rural population and relatively low livestock numbers, the latter being due to tropical farming prevailing over pastoralism. The western states, with their high representation of Afrotropical butterflies, are the most developed in India [112], and their semi-arid character is suitable for pastoral land use; these facts were reflected in the model’s predictor structure. For Palearctic species, the model terms reflected conditions of mountainous states with predominantly rural but relatively prosperous populations (cf. Sheratt [113]).

For all groups of predictors, the Himalayan state of Ladakh, with xero-mountain biomes and a distinct high mountain climate, displayed maximal species turnover, or dissimilarity from the rest of the Indian states. In the climatic models, the arid western states and the summer monsoon western states were also differentiated from the rest of peninsular India, but less distinctly than Ladakh.

## 5. Conclusions

The peculiar Indian geography, along with its climate, exerts dominant effects on butterfly species richness. Species-richness patterns copy the position of well-established biodiversity hotspots, mainly the northeast, Western Himalayan and the Western Ghats, and suggest that the northeastern states of Mizoram and Tripura remain under-sampled at present. Another under-sampled state is Telangana, situated in the relatively species-poor peninsular India. From a global conservation perspective, the federal states within the above hotspots are of the highest priority. However, regarding Indian fauna, the prevailingly Palearctic Ladakh and the arid northwestern states are unique and deserve increased attention. While the richness of all and Oriental elements flourish in densely forested states, the regionally interesting Afrotropical elements abound in the little-appreciated seasonal savannahs of western states. As with the current socioeconomic conditions, the related regionally specific land use patterns coexist with the amazing richness of butterfly species; future development should respect such regional features as basic landscape structures (e.g., the current rural landscape in many of the savannah states likely host similar insect communities as an original savannah biome). These observations justify the utility of using checklists for unequally sized administrative units, both for analyzing large-scale biogeography patterns [18] and for future comparisons of fauna change [20]. This, however, does not downplay the importance of having more detailed record-keeping, which may eventually lead to the production of equal grids atlases [1].

## Figures and Tables

**Figure 1 insects-14-00549-f001:**
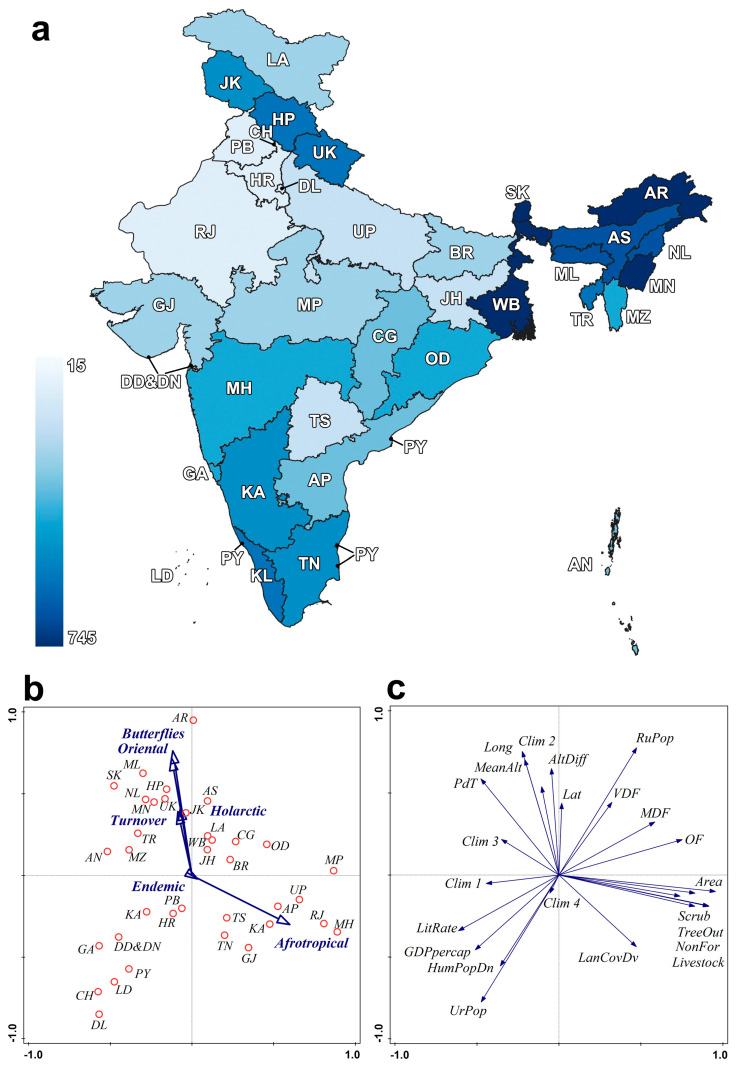
(**a**) Map of Indian federal states, with state names, abbreviations and tone of blue indicating numbers of butterfly species. (**b**) Ordination scatterplot of Indian states, obtained from a principal component analysis organizing the states according to values of predictors used in the regression models (eigenvalues 1–4: 0.336, 0.197, 0.135, 0.076, explained variation 35.0%); the darts indicating per-state butterfly numbers were entered as supplementary variables, not influencing the positions of the states. (**c**) PCA scatterplot of the predictors used in the regression models. Legend to Figure 1: Indian states with respective butterfly species richness: AN—Andaman and Nicobar (242), AP—Andhra Pradesh (229), AR—Arunachal Pradesh (745), AS—Assam (686), BR—Bihar (174), CG—Chhattisgarh (203), CH—Chandigarh (100), DD&DN—Daman and Diu and Dadra and Nagar Haveli (37), DL—Delhi (106), GA—Goa (275), GJ—Gujarat (200), HP—Himachal Pradesh (386), HR—Haryana (118), JH—Jharkhand (162), JK—Jammu and Kashmir (315), KA—Karnataka (328), KL—Kerala (330), LA—Ladakh (180), LD—Lakshadweep (15), MH—Maharashtra (284), ML—Meghalaya (712), MN—Manipur (730), MP—Madhya Pradesh (184), MZ—Mizoram (276), NL—Nagaland (643), OD—Odisha (246), PB—Punjab (129), PY—Puducherry (62), RJ—Rajasthan (124), SK—Sikkim (729), TN—Tamil Nadu (329), TR—Tripura (349), TS—Telangana (151), UK—Uttarakhand (518), UP—Uttar Pradesh (153), WB—West Bengal (731). Variables in panel (**c**): AltDiff—altitude difference, Clim1–4—composite climatic variables obtained by PCA analysis of 19 bioclimatic variables, GDPpercap—GDP per capita, HumPopDn—human population density, LanCovDv—land cover diversity, Lat—latitude, LitRate—literacy rate, Long—longitude, MeanAlt—average altitude, MDF—moderate dense forest, NonFor—non forest, OF—open forest, PdT—precipitation/temperature ratio, RuPop—rural population, TreeOut—scattered tree patch, UrPop—urban population, VDF—very dense forest.

**Figure 2 insects-14-00549-f002:**
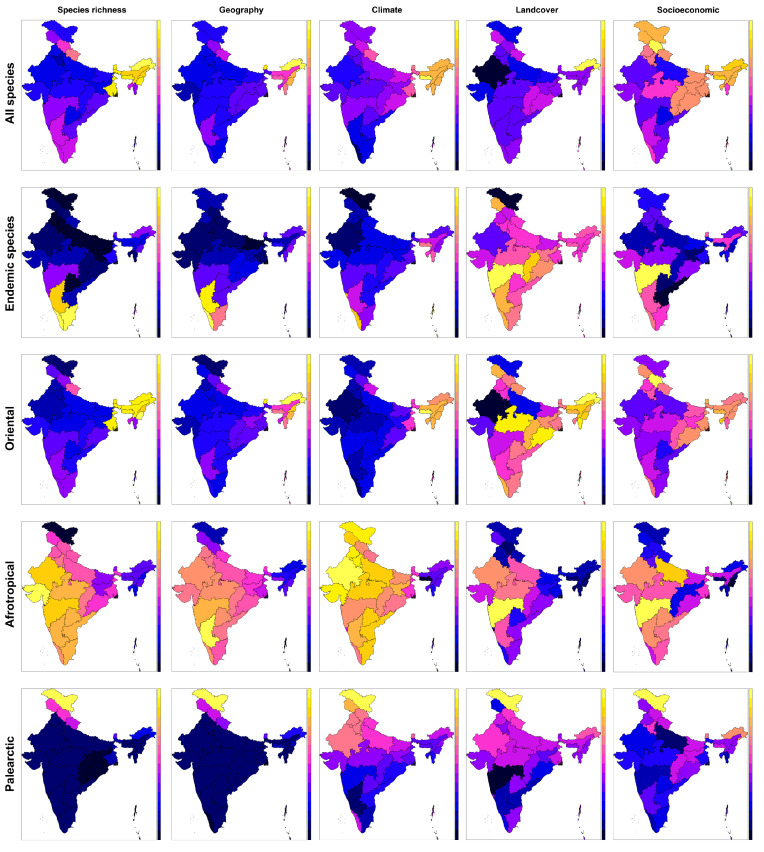
Maps showing the richness of all, endemic, Oriental, Afrotropical and Palearctic elements in 36 federal states and territories of the Republic of India (raw data: left column), and predictions of the regression models (four right columns), explaining the left column patterns by sets of geographic, climatic, land cover and socioeconomic predictors. See Table 3 and Table 4 for the models’ terms and related statistics. The color scales run from highest values (light yellow) to lowest values (dark blue).

**Figure 3 insects-14-00549-f003:**
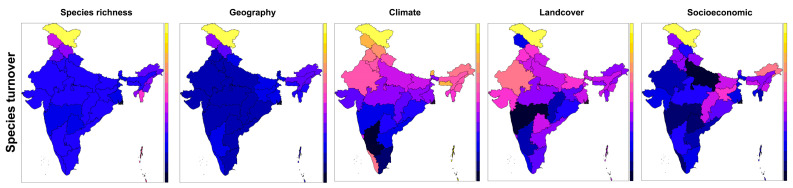
Maps showing the turnover of butterfly species compositions in 36 federal states and territories of the Republic of India (raw data: left), and predictions of the regression models, explaining the turnover pattern by sets of geography, climate, land covers and socioeconomic predictors. See Table 3 for the models’ terms and related statistics.

**Table 1 insects-14-00549-t001:** Overview of predictors describing the geography, land cover, socioeconomic conditions and climate in 36 federal states and union territories of the Republic of India, used for modeling butterfly species richness in the states.

Variable	Mean ± SD	Median	Range	Variable	Mean ± SD	Median	Range
**Geographic**				**Climatic ***			
Area (km^2^)	91,331 ± 95,597.5	55,149	30–342,239	Bio1	223.90 ± 71.90	253.14	−37.78–283.88
Altitude difference (m)	1549.14 ± 1781.91	1009.72	0–6117.05	Bio2	106.82 ± 21.71	104.81	54.67–144.44
Mean altitude	752.7 ± 1057.87	325.0	5–4730.5	Bio3	45.50 ± 8.23	44.65	28.27–63.40
Latitude	22.97 ± 6.97	23.82	8.28–34.96	Bio4	4268.31 ± 2045.55	4045.79	631.91–9355.72
Longitude	81.74 ± 7.32	79.11	71.57–94.67	Bio5	337.51 ± 69.41	356.91	155.58–412.48
**Land covers**				Bio6	93.30 ± 95.27	102.65	−228.58–231.67
Very Dense Forest (km^2^)	2771.61 ± 3983.71	1447.97	0–21,058.4	Bio7	244.20 ± 74.88	250.41	87–384.15
Moderately Dense Forest (km^2^)	8524.71 ± 9364.15	5463.5	13.51–34,209	Bio8	247.66 ± 61.63	266.60	17.46–300.61
Open Forest (km^2^)	8531.12 ± 7991.45	8638.46	8.01–36,618.6	Bio9	193.67 ± 84.43	214.08	−95.38–276.12
Scrub (km^2^)	1292.75 ± 2047.64	309.11	0–8276.09	Bio10	273.78 ± 60.73	296.40	78.88–329.25
Non-Forest (km^2^)	70,210.7 ± 84,277.8	41179	2.9–320,775	Bio11	164.38 ± 91.18	182.25	−161.28–267.67
Scattered tree cover (km^2^)	2659.67 ± 3059.24	1281.5	0.05–12,108	Bio12	1477.74 ± 822.24	1208.31	152.72–3909.38
Landcover diversity	1.51 ± 0.22	1.50	1.04–1.96	Bio13	382.17 ± 217.29	346.10	27.08–1040.23
**Socioeconomic**				Bio14	6.13 ± 5.50	4.07	0–21.91
Livestock (million)	15.72 ± 19.57	6.48	0.02–68.71	Bio15	104.32 ± 25.95	100.51	53.68–152.86
Human population density (km^−2^)	1024.32 ± 2357.16	313.5	4.6–11,320	Bio16	941.62 ± 510.03	860.59	66.49–2427.74
Urban population (million)	37.17 ± 21.77	29.59	10.03–97.5	Bio17	36.31 ± 27.55	29.25	0.90–108.73
Rural population (million)	62.82 ± 21.78	70.41	2.5–89.97	Bio18	472.62 ± 418.34	317.11	46.16–1984.83
Literacy rate (%)	77.15 ± 9.34	77.1	50.96–94	Bio19	124.06 ± 216.07	41.97	1.41–1136.12
GDP per Capita ($)	2411.54 ± 1231.91	2411.54	588.8–5695.3	Bio12/Bio1	6.93 ± 4.73	5.33	−4.04–19.13

* Bioclimatic variables: [Bio]1: annual mean temperature; 2: mean diurnal temp. range; 3: isothermality; 4: temp. seasonality; 5: max temp. of warmest month; 6: min temp. of coldest month; 7: temp. annual range; 8: mean temp. of wettest quarter; 9: mean temp. of driest quarter; 10: mean temp. of warmest quarter; 11: mean temp. of coldest quarter; 12: annual precipitation; 13: mean prec. of wettest month; 14: prec. of driest month; 15: prec. seasonality (coefficient of variation); 16: prec. of wettest quarter; 17: prec. of driest quarter; 18: prec. of warmest quarter; 19: prec. of coldest quarter.

**Table 2 insects-14-00549-t002:** Results of univariate models used to test the basic biogeographic predictions for butterfly fauna of Indian states (generalized linear models, dependent and independent variables transformed to 0 mean and unit variance). All the fitted models have 1, 34 degrees of freedom and are compared with the y~+1 null model, with null deviance = 35, 35 df, AIC = 105.1.

	Coefficient	D^2^	Deviance	AIC
**Area**				
Richness	−0.187	3.4%	33.8	105.9
Endemic	+0.053	0.3%	34.9	107
Oriental	−0.204	4.0%	33.6	105.6
Afrotropical	+0.598	35.7%	22.5	91.2
Palearctic	−0.011	0.0%	35	107.1
**Land cover diversity**				
Richness	−0.491	24.0%	26.6	97.2
Endemic	−0.300	8.9%	31.9	103.8
Oriental	−0.525	27.4%	25.4	95.6
Afrotropical	+0.324	10.6%	31.3	103.2
Palearctic	+0.130	1.7%	34.4	106.5
**Precipitation/Temperature ratio**				
Richness	+0.680	46.3%	18.8	84.8
Endemic	+0.210	4.3%	33.5	105.5
Oriental	+0.697	48.6%	18.0	83.2
Afrotropical	−0.297	8.9%	31.9	103.8
Palearctic	−0.005	0.0%	35.0	107.1
**Altitude difference**				
Richness	+0.500	24.9%	26.3	96.8
Endemic	+0.078	0.6%	34.8	106.9
Oriental	+0.383	14.6%	29.9	101.5
Afrotropical	−0.011	1.1%	34.6	106.7
Palearctic	+0.853	73.0%	9.45	60.1
**Latitude**				
Richness	+0.225	5.1%	33.2	105.3
Endemic	−0.510	26.0%	25.9	96.4
Oriental	+0.146	2.0%	34.3	106.4
Afrotropical	+0.082	0.6%	34.8	106.9
Palearctic	+0.568	32.3%	23.7	93.15

**Table 3 insects-14-00549-t003:** Overview of the multivariate models for butterfly species richness, endemic species and species turnover of Indian states, describing the physical geography, climate, land covers and socioeconomic variables (generalized linear models, both dependent and independent variables transformed to 0 mean and unit variance). All respective models were compared with a null model y~+1, with null deviance = 35.0, 35 df, AIC = 105.1.

	Species Richness					Endemic Species					Species Turnover				
	Model *	Dev	D^2^	df	AIC	Model	Dev	D^2^	df	AIC	Model	Dev	D^2^	df	AIC
**Geography**	+3.17 Long +0.83 Long^2^ +1.18 meanAlt −0.35 Lat −0.65 meanAlt:Lat	10.4	70.3	30	71.5	+0.88 Long +0.79 Long^2^ +2.63 meanAlt −12.61 Lat +6.07 Lat^2^ −18.37 meanAlt:Lat +5.74 meanAlt:Lat^2^	7.8	77.7	28	65.3	+2.70 Long +0.70 Long^2^ +0.35 meanAlt −1.40 Lat +2.62 Lat^2^ −3.53 meanAlt:Lat +3.88 meanAlt:Lat^2^	6.8	80.6	28	60.3
**Climate**	+0.73 Clim2	15.9	54.5	34	78.8	+0.43 Clim1	28.3	19.1	34	99.5	+0.23 Clim2 +0.56 Clim3	21.9	37.4	33	92.3
**Land covers**	+ 0.37 VDF −0.31 NonFor	27.8	20.5	33	100.9	−0.81 NonFor +0.91 Treeout	29.1	16.9	33	102.5	+0.97 NonFor −1.06 Treeout	27.0	22.9	33	99.9
**Socioeconomics**	+0.63 RuPop −0.43 Livestock	20.8	40.6	33	90.5	+0.34 LitRate −0.27 HumPopDens	30.2	13.7	33	103.8	−0.51 Livestock −0.56 LitRate	24.6	29.7	33	96.5

* See legend to Figure 1 for abbreviations of predictors.

**Table 4 insects-14-00549-t004:** Overview of the multivariate models for biogeographic elements (Afrotropical, Oriental and Palearctic) in butterfly fauna of Indian states, describing the physical geographic, climatic, land cover and socioeconomic variables (generalized linear models, both dependent and independent variables transformed to 0 mean and unit variance). All the fitted models have 1, 34 degrees of freedom and are compared with the null model, in which y~+1, with null deviance = 35, 35 df, AIC = 105.1.

	Afrotropical					Oriental					Palearctic				
	Model *	Dev	D^2^	df	AIC	Model	Dev	D^2^	df	AIC	Model	Dev	D^2^	df	AIC
**Geography**	−3.72 Long −0.92 Long^2^ +1.36 meanAlt −4.61 Lat +0.54 Lat^2^ −10.2 meanAlt:Lat +2.71 meanAlt:Lat^2^	13.8	60.6	28	85.7	+3.35 Long +0.91 Long^2^ +1.14 meanAlt −0.39 Lat −0.71 meanAlt:Lat	9.9	71.7	30	69.6	−0.30 Long −0.24 Long^2^ +0.25 meanAlt +0.32 Lat +0.49 meanAlt:Lat	0.93	97.3	30	−15.6
**Climate**	−0.32 Clim1 −0.41 Clim2	25.3	27.7	33	97.4	+0.18 Clim1 +0.73 Clim2 −0.21 Clim4	12.9	63.1	32	75.4	−0.37 Clim1 +0.53 Clim3	20.1	42.6	33	89.2
**Land covers**	+0.69 Treeout	18.5	47.1	34	84.3	+0.37 VDF −0.33 NonFor	27.4	21.7	33	100.4	+0.34 MDF −0.30 Scrub +1.25 NonFor −1.29 Treeout	24.7	29.4	31	100.5
**Socioeconomics**	+0.71 Livestock +0.33 GDPpercap	19.4	44.6	33	87.8	−0.30 Livestock +0.73 RuPop −0.29 LitRate	21.1	39.7	32	93.0	−0.51 Livestock +0.32 RuPop −0.58 LitRate +0.36 GDPpercap	20.4	41.7	31	93.7

* See legend to Figure 1 for abbreviations of predictors.

## Data Availability

All of the data are provided in the Appendix A of the manuscript.

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
