# Peer review of "Geography of Indian Butterflies: Patterns Revealed by Checklists of Federal States"

_insects, 2023, doi:10.3390/insects14060549_

Round 1

Reviewer 1 Report

The manuscript entitled “Geography of Indian butterflies: Patterns revealed by checklists of federal states” highlights the use of butterflies as indicators for analyzing biogeography patterns at both global and regional scales. The findings of the study revealed that factors such as area, land cover diversity, and latitude did not significantly impact species richness. However, topographic diversity and the ratio of precipitation to temperature were identified as positive predictors of butterfly species richness.

Overall, this research provides valuable insights into the factors influencing butterfly species richness in India and highlights the conservation significance of specific regions within the country.

I found this to be an excellently written study that I really enjoyed reading. The work is well-designed and carried out experimentally, and the analyses and sampling size are robust. Regarding the subject matter of the study, the writers have a thorough comprehension of biogeographical and ecological requirements of their studied organisms and the exceptional topography of their study region. In addition, the authors are aware of up-to-date literature and have a deep scientific knowledge with regards to the paper scope.

This study is an outstanding contribution to the existing literature on biogeographical patterns of butterflies in a biodiversity hotspot because it provides data on Indian butterflies that are less studied compared to the butterfly fauna that can be found in other countries and continents. It also gives access to an amazing dataset of 1379 butterfly species (Supplementary table S1) recorded in the less study federal states of the Republic of India. As the authors suggest and I fully agree, this study “may be used as a baseline for future conservationists, and current biogeographers”.

I provide below only some minor suggestions, hoping to be of use of the improvement of the paper.

Line 129: Please delete “then”

Line 131: Please replace “for which we used the” with “using”

Line 145: I am not sure what “CV” stands for.

Table 1: I could not find what Bio1-Bio19 represent.

Figure 1: Again here, I am not sure where I can find the explanation of the climatic variables you are using in your analysis.

Figure 2: it’s a bit confusing what species richness x species richness panel shows here. But perhaps I’m missing something.

Table 3: I think you need to define the coded predictors here as well.

Author Response

COMMENTS OF REVIWER 1

#The review starts with very positive assessment, which we, of course, deeply appreciate. Then, it goes to specific comments.

Line 129: Please delete “then”

#Done.

Line 131: Please replace “for which we used the” with “using”

#Done.

Line 145: I am not sure what “CV” stands for.

#It stood for coefficient of variation, changed.

Table 1: I could not find what Bio1-Bio19 represent.

#These were standard abbreviations of bioclimatic variables from Karger et al., Sci. Data 2017, 4, 170122. We now added their names into extended figure legend.

Figure 1: Again here, I am not sure where I can find the explanation of the climatic variables you are using in your analysis.

#We added the names to the Table 1 legend, and also prepared a new supplementary material, where they are also named.

Figure 2: it’s a bit confusing what species richness x species richness panel shows here. But perhaps I’m missing something.

#You are right that this was confusing. What we meant were “all species” at the y-axis of the graphs. Corrected. We also removed the numbers of the colour scales, as they were not readable anyway and replaced them in explanations in figure legends, see lines 280-86.

Table 3: I think you need to define the coded predictors here as well.

#This would intolerably expand the table, so we refer readers to legend to Figure 1. But thank you a lot for this suggestion.

Reviewer 2 Report

This paper addresses the biogeography of butterflies in India, testing various drivers including habitat filtering and dispersal barriers. The data that are used are species lists for all the states that make up India. This is a worthwhile endeavor and nice graphical and statistical results are presented.

The MS has many shortcomings though that need to be addressed.

Most importantly, the announced statistical methods and what we find in the results are partly disconnected. We get results of tests that were not announced or inadequately described.

L158. ‘We ran single-term regression for all potential predictors, plus their second-order polynomials and first-order interactions with all other predictors. We then constructed a combined model containing all predictors..’. I didn’t understand this very well. I asked ChatGPT to explain it to me. It seems that you started with a full model with all the predictors and their quadratic terms, + interactions (why is this called ‘single-term’?). This seems to risk overfitting of the data given the limited number of states. Then you reduced complexity by excluding terms based on AIC (should be cAIC?) stepwise. Have you tried the dredge function in R (package MuMIn)? However, later seeing Tables 3 and 4 I see that you had separate models for different categories of predictors. Wouldn’t it make sense to (also) mix categories and look at what best explains the patterns?

Could there be a problem of spatial autocorrelation? (Neighboring states being similar) How would this affect the results? Can you include correlations between predictors as a table?

It seems that the PCA goes into the regression, so it would make more sense to describe this first. What program was used for PCA?

India is not well suited to distinguishing between latitudinal effects and peninsular effects as they are colinear. As long as you are in the tropics, the diversity is likely high unless it is remote from big chunks of the wet tropics (e.g. western Ghats being far away from South-East Asia and Africa)). I would include the ‘distance to SE Asia’ forest block as a factor. Isn’t ‘available energy’ one of the explanations for the ‘poleward decrease of species richness’, not a separate pattern?

Generally, a political boundary is not a sensible scientific boundary. This limitation to India should be justified by e.g. the lack of such comparable data for neighbouring countries.

The use of socio-economic factors as a predictor is questionable. It assumes that human activities have led to significant extinctions of butterflies in states. Most likely almost all original butterfly species still hang on in remaining (protected) wild areas and these are usually sampled by the naturalists. At this scale, these parameters seem less relevant. For large mammals, the situation may be different.

I’d like to see the contrast between forest and savannah species.

The introduction starts immediately with methodology and has other reversals of the order of information. Go from the general issue (global patterns+ include more literature from other parts of the world that address similar questions) to the specific issue (pattern in India) to gaps in the knowledge of such patterns in Indian butterflies to methods to study these patterns. Leading up to the specific questions and approaches used in the paper. History pops up at various places in the introduction, try to keep it together.

There are numerous mistakes such as ‘land covers’ (should be land cover), ‘mountainous forests’ (should be ‘forested mountains’), within its borders. I note a few below. Please use a tool like Grammarly (its free) and careful editing by all authors before submission of MSs.

I trust that everybody knows that India is a large tropical country. Later called the ‘Republic of India’. Be consistent.

L97 add bulk in tropical regions

L100 can relationships be valid? They can exist or not, or be more or less strong.

L100-104 long and unnecessarily complex sentence.

L 108 (but see Karanth [58], eBird [59]), should be clarified as (see for mammals and birds…). It would be good to describe the results of these studies briefly in the Introduction, as they also provide hypotheses for butterflies.

L116 updated with

Table 1 is very boring. What is the use? Moreover, The Bio1 to Bio19 are meaningless for readers. Better have an appendix with the data per state with perhaps an average included per variable.

L175 don’t give 1 sentence its own paragraph.

L181 Reformulate ‘ranking of species-richest states’.

L183 simply give % for each.

L185 ‘effectively refuting a poleward species richness decrease, while not supporting a decrease towards the tip of the peninsula’. Since latitude and peninsular width are strongly correlated, it is not possible to distinguish the two. Possibly they simply cancel each other out here. Anyway, this is discussion, better keep it for Discussion section.

Figure 1: is the tone of blue the number of species? Not said in legend. The ordination scatterplot and PCA are not announced and detailed in the methods. The words inside the graph are very small and not easy to understand. Abbreviations should be explained in the figure legend, not the main text! In itself, this approach is probably quite promising. It needs to be detailed in methods and results (stats on the results). Maybe you could use the package vegan to perform NMDS or RDA and use the envfit function (perhaps you did). Use e.g. PERMANOVA to test predictors of the butterfly communities. It is quite easy to do.

Table 2 is also not announced in the methods. The organization is strange. I would go by response variable and then list predictors ordered by AIC. Otherwise useful table.

Figure 2 top left panel left label is the same as the top label. It should be ‘all species’.  This should be the same as the left panel in Figure 1, so the left panel in Figure 1 can be removed from MS.

The final sentence ‘These observations justify the utility of our comparisons, which may be used as a baseline for future conservationists, and current biogeographers.’ Is very frivolous.

use a tool like Grammarly

Author Response

COMMENTS OF REVIEWER 2

This paper addresses the biogeography of butterflies in India, testing various drivers including habitat filtering and dispersal barriers. The data that are used are species lists for all the states that make up India. This is a worthwhile endeavor and nice graphical and statistical results are presented.

The MS has many shortcomings though that need to be addressed.

Most importantly, the announced statistical methods and what we find in the results are partly disconnected. We get results of tests that were not announced or inadequately described.

L158. ‘We ran single-term regression for all potential predictors, plus their second-order polynomials and first-order interactions with all other predictors. We then constructed a combined model containing all predictors..’. I didn’t understand this very well. I asked ChatGPT to explain it to me. It seems that you started with a full model with all the predictors and their quadratic terms, + interactions (why is this called ‘single-term’?). This seems to risk overfitting of the data given the limited number of states. Then you reduced complexity by excluding terms based on AIC (should be cAIC?) stepwise. Have you tried the dredge function in R (package MuMIn)? However, later seeing Tables 3 and 4 I see that you had separate models for different categories of predictors. Wouldn’t it make sense to (also) mix categories and look at what best explains the patterns?

#This comment addresses several issues, some of which may be just misunderstandings. To begin with, there are potentially countless ways, how to produce regression models of factors influencing species richness. In our case, understanding that this is a basic “exploratory” analysis of patterns, and that we work with rather low numbers of freedom (defined by the number of federal states), we proceeded by: 1) selecting predictors (in linear or quadratic forms) having separate effect, measured by decreasing null model AIC; 2) constructing a saturated model from those predictors; and 3) manually eliminating this model via backward-elimination procedure; while 4) sequentially checking for nominal (P<0.05) significance of all model terms in each step. We fully understand that a more fancy ways would be possible, but this would hardly add much value to the results.

Regarding the categories of predictors, and mixing of the categories: Such mixing is exactly what we wanted to avoid, understanding that the very first question a geographer asks, when seeing a pattern, such as species richness distribution, are questions such as, “Is this due to geography, or climate? What is the role of land covers? Is this somehow connected to human activity?”, and so on. Of course, more complex questions could be asked, and addressed by mixing predictors, but the results would lose a beauty of simplicity, which we believe all scientists should value most. 

Could there be a problem of spatial autocorrelation? (Neighboring states being similar) How would this affect the results? Can you include correlations between predictors as a table?

#Spatial autocorrelation is always present in analyses of spatial data, but not always represents a problem. It is a big trouble, if a researcher seeks for patterns or experimental results independent on spatial position of samples or treatments etc., for instance an effect of reserve size of species richness, and some of the reserves are situated in species-rich and some in species-poor region. If one studies the effects of the spatial position, typically geography, the problem becomes the result by itself, which is our case. By showing the overwhelming geography effect, apparent not only in the explained variations, but also in the high values of regression coefficients (steepness of slopes), we are in fact showing that the species richness patterns are extraordinarily spatially correlated. In less spatially dependent data, one would proceed by retaining the spatial terms in the models, and modeling the effects of other variables. This was not doable in our case, however, because the geography, i.e. the spatial terms, explained almost entire explicable variation.

Regarding the correlations among predictors, these are partly visible from Figure1c, but we agree that presenting such results is always useful, and present them as Supplementary table 3 (see also lines 146-7).

It seems that the PCA goes into the regression, so it would make more sense to describe this first. What program was used for PCA?

#The program is presented at line 147-8, and again 178-9. We in fact used two PCA analyses, one for extracting the composite climatic variables Clim1–4 (lines 178-86), the other to visualize the interrelations among predictors (Figure 1). For the former, we now prepared Supplementary table S 3, presenting results of the bioclimatic PCA.

India is not well suited to distinguishing between latitudinal effects and peninsular effects as they are colinear. As long as you are in the tropics, the diversity is likely high unless it is remote from big chunks of the wet tropics (e.g. western Ghats being far away from South-East Asia and Africa). I would include the ‘distance to SE Asia’ forest block as a factor. Isn’t ‘available energy’ one of the explanations for the ‘poleward decrease of species richness’, not a separate pattern?

#Thank you for this comment. You are certainly right that India is poorly suited for peninsular effect testing due to possible confusion with latitudinal gradient, but thinking about this issue, very few of the Earth’s major peninsulas are, because most of them run in similar directions (Florida and Baja California, the three Mediterranean peninsulas, Malay peninsula, even the southernmost tips of Africa and South America). Still, in a simplistic view, species richness should increase towards the Equator, and the absence of such increase can be viewed as an indication of the peninsular effect presence. Of course, the “equatorial explanation” is no explanation at all, it is rather the distance towards wet tropics, which tend to be distributed near the Equator (in Africa, South America, and the “Farther Indies”)  - but not always so (which is the case of Republic of India). However, we are not aware of an explicit hypothesis, which would re-state the latitudinal effect in terms of “chunks of wet tropics”, but we are aware of the “available energy” formulations, which basically state the same. Hence, “available energy” (one of the strongest predictors in our results) is basically stating the same (the ratio we used is highest in the wet tropics).

Generally, a political boundary is not a sensible scientific boundary. This limitation to India should be justified by e.g. the lack of such comparable data for neighbouring countries.

# We added this justification, see lines 401-402.

The use of socio-economic factors as a predictor is questionable. It assumes that human activities have led to significant extinctions of butterflies in states. Most likely almost all original butterfly species still hang on in remaining (protected) wild areas and these are usually sampled by the naturalists. At this scale, these parameters seem less relevant. For large mammals, the situation may be different.

#We agree that relating socioeconomic factors and biodiversity is always questionable, and fully endorse your explanation that even in the most human pressured regions, some species will be surviving in “in remaining (protected) wild areas”, and so, current indicators of human pressure will unlikely affect current species richness. However, such results can be used in historical (or future) comparisons, as change of species richness, extinction rates, threat rates and other such “dynamic” indicators can be tightly related to socioeconomic factors, as shown, e.g., for European butterflies by Konvicka et al., Global Ecol Biogeogr, 2006, 15, 86-92, or recently by Gosselin and Callois, Anthropocene 2021, 35, for multiple taxa. Any future use, however, needs assessment of the current situation, and hence we opt to retain these results in the manuscript. 

I’d like to see the contrast between forest and savannah species.

#Almost all species categorized as “Afrotropical species” inhabit savannahs. Such species are for instance Belenois aurota, or several species of Azanus and Taurucus (but not all of them are Afrotropical!), whereas vast majority of Oriental species are known as forest species. Here, the contrast is well visible for instance in Fig. 2, their distribution patterns are rather opposite. We added relevant sentence with examples to the manuscript, see lines 506-508.

The introduction starts immediately with methodology and has other reversals of the order of information. Go from the general issue (global patterns+ include more literature from other parts of the world that address similar questions) to the specific issue (pattern in India) to gaps in the knowledge of such patterns in Indian butterflies to methods to study these patterns. Leading up to the specific questions and approaches used in the paper. History pops up at various places in the introduction, try to keep it together.

#We understand your comments, but beg to disagree. The literature on global patterns from other parts of World is immense and each of the patterns (equatorial increase, effect of heterogeneity, effect of energy…), is covered by a library. Not even talking about their combinations, various modelling studies, etc. This creates the near-absurd situation, that properly introducing all the patterns (and their manifestations for various groups of organisms, and exceptions for various regions) would result in either intolerably long Introductory section, or in stating a few selected trivia. Plus, there was the more focused literature on India, including the historical one – by itself large, we had to do some weeding, but still, we considered it more important to present the Indian biogeography + butterflies in the Introduction, even if scarifying a deeper overview of macroecology literature. We also reasoned that for most of readers interested in this particular paper, the basic macroecology literature is rather familiar – in contrast to basic Indian butterflies’ literature.

Hence, we went directly to the issue. It si not “methodology” as such, but explanation why per-state checklists can be useful in absence of sources such as grid-based atlases, which represent a standard in macroecology literature, despite having issues of their own (generated, e.g., by unequal sampling). From this, we go into main testable patterns, and Indian peculiarities. Please, trust us, that writing the Introduction in a more standard way would produce something which would be much longer, and much less readable.

There are numerous mistakes such as ‘land covers’ (should be land cover), ‘mountainous forests’ (should be ‘forested mountains’), within its borders. I note a few below. Please use a tool like Grammarly (its free) and careful editing by all authors before submission of MSs.

#The paper had been thoroughly checked by an English native speaker. Some of the mistakes were genuine, and we corrected them, but please, some of them are not: e.g., “mountainous rainforest region” is exactly what it is, the region with mountains and rainforests.

I trust that everybody knows that India is a large tropical country. Later called the ‘Republic of India’. Be consistent.

#Well, let us retain this truism (large + tropical) here, as it is pivotal for carrying out the analyses. But we changed it to RoI, to be consistent, see line 62.

L97 add bulk in tropical regions
#Corrected.

L100 can relationships be valid? They can exist or not, or be more or less strong.

#You are right. We changed the term “relationship” into “patterns”, as all the effects tested (area, latitude, etc.) are indeed rather patterns – observed situations – than relationships, as a relationship implies a relation (cause-effect situation).

L100-104 long and unnecessarily complex sentence.

#We split the sentence into two, and simplified it, see line 101-105.

L 108 (but see Karanth [58], eBird [59]), should be clarified as (see for mammals and birds…). It would be good to describe the results of these studies briefly in the Introduction, as they also provide hypotheses for butterflies.

#We changed the section according to your advice, see lines 109-114.

L116 updated with

#Changed as advised.

Table 1 is very boring. What is the use? Moreover, The Bio1 to Bio19 are meaningless for readers. Better have an appendix with the data per state with perhaps an average included per variable.

#The use of the table is to present all the predictor variables, which we believe is a good practice in pattern-searching studies, as it, if anything else, it shows statistical distributions of the variables, their means, medians, etc.

L175 don’t give 1 sentence its own paragraph.

#This is an important paragraph, because it informs on the software used, and cannot really be merged with the previous text.

L181 Reformulate ‘ranking of species-richest states’.

#Changed to “most speciose states”.

L183 simply give % for each.

#Changed as suggested.

L185 ‘effectively refuting a poleward species richness decrease, while not supporting a decrease towards the tip of the peninsula’. Since latitude and peninsular width are strongly correlated, it is not possible to distinguish the two. Possibly they simply cancel each other out here. Anyway, this is discussion, better keep it for Discussion section.

#You are right that they may cancel each other, which we explain in more detail now (lines 215-18). However, as you observed elsewhere, there is also the effect of wet vs. dry tropics, which we address in Discussion.

Figure 1: is the tone of blue the number of species? Not said in legend. The ordination scatterplot and PCA are not announced and detailed in the methods. The words inside the graph are very small and not easy to understand. Abbreviations should be explained in the figure legend, not the main text! In itself, this approach is probably quite promising. It needs to be detailed in methods and results (stats on the results). Maybe you could use the package vegan to perform NMDS or RDA and use the envfit function (perhaps you did). Use e.g. PERMANOVA to test predictors of the butterfly communities. It is quite easy to do.

#We added explanation of tones of blue, lines 258-9, and introduced the PCA of predictors at appropriate place. We also added the abbreviations of predictors.

Table 2 is also not announced in the methods. The organization is strange. I would go by response variable and then list predictors ordered by AIC. Otherwise useful table.

#It is now announced, see lines 166-68. Regarding the organization, as the predictors are focal for the table, we opted organizing it by predictors.

Figure 2 top left panel left label is the same as the top label. It should be ‘all species’.  This should be the same as the left panel in Figure 1, so the left panel in Figure 1 can be removed from MS.

#Changed to all species, thank you for this correction. Otherwise, the top-left of Figure 2 is the same map as in Figure 1, but Figure 1 also introduces the Indian states for the reader and shows scale with true species numbers. Its repetition in Figure 2 does not occupy so much space, and allows direct visual comparison with the predicted numbers, and so we would like to retain is there.

The final sentence ‘These observations justify the utility of our comparisons, which may be used as a baseline for future conservationists, and current biogeographers.’ Is very frivolous.

#Interestingly, the other two reviewers praised the sentence. We, however, reformulated into a more civic tone.

Reviewer 3 Report

In their article Das et al. provide a state-level analysis of the distribution and potential drivers of species richness and endemism in India based on local checklists. They show that species richness is highest in northwestern states and the number of endemics is highest in Southern India. Testing for a broad range of common drivers they find that tree cover, topological diversity and certain socioeconomical as well as climatic aspects predict butterfly diversity in India at the state-level. I find the article overall well-written and the analysis mostly adequate. However, to fully grasp the reliability of the results I recommend the inclusion of more methodological details and the exploration of potential caveats. Also, the discussion needs to address potential applications of such checklists beyond the scope of this study. In general, I think the article is a nice contribution to the field and provides a valuable resource as well as first step to explorations of butterfly diversity patterns in the tropics.

Major comments:

L26-27: I disagree with this statement. The authors should rather present their findings as an initial step towards understanding the distribution and diversity of butterflies and highlight that a comprehensive checklist for regions can facilitate data validation as well as integrative approaches for more fine-scaled assessments (see e.g. Pinkert et al. 2022 in GEB for arguments and Sandall et al. 2022 in JBI for another integrative approach).

L35-39: I am curious how the authors addressed the impact mountains (topographic diversity is not the same as altitude, for instance). In fact, mean altitude seems more important in the models (Table 3). Please clarify what you mean by “topographic diversity” in the abstract as the statement doesn not seem to be supported by you results.

L188: Please consider using the residuals of the relationship between e.g. species richness~area for further analysis or an area corrected measure as you do not seem to be interested in the effect of area per se and even not significant effects will influence your estimates in downstream analysis. In addition, I recommend a simple but more appropriate test for an area-effect using Spearman’s rank correlation instead of a linear regression.

L207: Please report PCA loadings and further information in a supplementary table to aid the interpretation of the PCA results.

Conclusions: Either in this section or in an additional paragraph in the discussion you should extend on discussing applications of your checklist-dataset that are relevant for further and potentially more detailed research on biodiversity pattern in butterflies. Please discuss limitations and extensions in the context of your work. The two publications of global checklists mentioned above and their comparisons with available occurrence records should be useful in this regard but there is also extensive on other use cases. Also think of ways the checklist can be easily updated or curated for further use.

Minor comments:

L16-20: The sentence is very long and should be split into at least 2 separate sentences to ease reading.

L20-24: Same for this sentence.

Table 3 and 4: These model results are essential to understand the predictions made in Figure 2 and 3 and I recommend that you at least include the proportion of explained variance in the figures. The quite large number and complex predictors (interactions with latitude etc.) to me highlight the likely issue of multicollinearity. To me it is very likely that you overfitted the models. Please report variance inflation factors for the predictors of the final models to confirm that this is not the case. Note that some of the predictors are not on separate rows which should be adapted to ease readability.

Figure 1: The text in these illustrative items is very hard to read and should be increases in size. Please also add the full names of (at least) the predictors to allow the reader to understand the figure independent from the manuscript text. I also highly recommend that you add similar Figures for the number of endemic species and species turnover as this is not provided though Fig 2 or 3 (where you present predictions not the raw data.

Figures 2 and 3: The tick marks of the scale are too small.

Supporting information: Note that temperature variables need to be divided by 10, which should not influence your results but needs to be adjusted in the table [e.g. 166 °C is not correct] and may influence your interpretations whenever you refer to exact values and coefficients (e.g. slopes an intercepts in model results). Please check throughout the manuscript and adapt.

Author Response

COMMENTS TO REVIEWER 3

In their article Das et al. provide a state-level analysis of the distribution and potential drivers of species richness and endemism in India based on local checklists. They show that species richness is highest in northwestern states and the number of endemics is highest in Southern India. Testing for a broad range of common drivers they find that tree cover, topological diversity and certain socioeconomical as well as climatic aspects predict butterfly diversity in India at the state-level. I find the article overall well-written and the analysis mostly adequate. However, to fully grasp the reliability of the results I recommend the inclusion of more methodological details and the exploration of potential caveats. Also, the discussion needs to address potential applications of such checklists beyond the scope of this study. In general, I think the article is a nice contribution to the field and provides a valuable resource as well as first step to explorations of butterfly diversity patterns in the tropics.

Major comments:

L26-27: I disagree with this statement. The authors should rather present their findings as an initial step towards understanding the distribution and diversity of butterflies and highlight that a comprehensive checklist for regions can facilitate data validation as well as integrative approaches for more fine-scaled assessments (see e.g. Pinkert et al. 2022 in GEB for arguments and Sandall et al. 2022 in JBI for another integrative approach).

#We changed the statement accordingly. The problem here is, that it is in part of Abstract called “Simple summary” by the Journal. The guidelines for this section specifically ask to avoid too complicated explanations and details, as it should be understandable for general public. This, of course, required some simplification.

L35-39: I am curious how the authors addressed the impact mountains (topographic diversity is not the same as altitude, for instance). In fact, mean altitude seems more important in the models (Table 3). Please clarify what you mean by “topographic diversity” in the abstract as the statement does not seem to be supported by you results.

#We fully agree that topographic diversity and altitude are different things, and that the former tends to be more important at smaller scales (e.g., single nature reserves), and the latter at broad scales. This is also the reason, why our manuscript asks two partly separate question, the first being to explore the “validity of selected basic macroecology patterns”, the second being “to explore factors putatively responsible for species richness”. Increasing species richness with topographic heterogeneity certainly is a well-known ecology pattern, but is not really “geographic” one. Hence, we used it in the simple analyses summarized in Table 2, but not in the more advanced models summarized in Table 3.

L188: Please consider using the residuals of the relationship between e.g. species richness~area for further analysis or an area corrected measure as you do not seem to be interested in the effect of area per se and even not significant effects will influence your estimates in downstream analysis. In addition, I recommend a simple but more appropriate test for an area-effect using Spearman’s rank correlation instead of a linear regression.

# Although we absolutely understand the overwhelming importance of area ~ richness relationship in spatial ecology, it did not have any effect at all in the regression analyses presented in Table 2 (see the AIC value, which is in fact higher than null model AIC). In such cases, working with residuals of area is clearly superfluous, as it inflates the unexplained deviation, instead of reducing it. The absence of area effect is clearly evident even in the Spearman rank correlation (Spearman’s r = 0.09, t = 0.52, p = 0.61), and so, we cannot really follow your advice. The absence of area effect is exactly the result of idiosyncrasies of Indian biogeography, with large and species poor states, such as Rajastan, and small and species rich states, such as Manipur.

L207: Please report PCA loadings and further information in a supplementary table to aid the interpretation of the PCA results.

#The information is in the Figure 1 legend.

Conclusions: Either in this section or in an additional paragraph in the discussion you should extend on discussing applications of your checklist-dataset that are relevant for further and potentially more detailed research on biodiversity pattern in butterflies. Please discuss limitations and extensions in the context of your work. The two publications of global checklists mentioned above and their comparisons with available occurrence records should be useful in this regard but there is also extensive on other use cases. Also think of ways the checklist can be easily updated or curated for further use.

#We expanded the conclusion according to your suggestion, see lines 548-551.

Minor comments:

L16-20: The sentence is very long and should be split into at least 2 separate sentences to ease reading.

#Split into two sentences, see line 17.

L20-24: Same for this sentence.

#Also split into two sentences, thank you for this suggestion.

Table 3 and 4: These model results are essential to understand the predictions made in Figure 2 and 3 and I recommend that you at least include the proportion of explained variance in the figures. The quite large number and complex predictors (interactions with latitude etc.) to me highlight the likely issue of multicollinearity. To me it is very likely that you overfitted the models. Please report variance inflation factors for the predictors of the final models to confirm that this is not the case. Note that some of the predictors are not on separate rows which should be adapted to ease readability.

#Reporting explained variance both in the tables and figures would be redundant, which the editors do not like, as they are already in the tables. Regarding the multicollinearity issue, by looking at the PCA diagram in Figure 1, some of the predictors, such as Human population density and Urban population, are indeed closely correlated, but notice, that such intercorrelated predictors never entered the same model together. Regarding overfitting, we were aware of this problem, and checked all the resulting models for residuals vs. fitted values, and standardized residuals vs. Quantiles, and there never was the issue. Note, also, that even in the most complex models (e.g., species richness ~ geography), the residual degrees of freedom are still reasonably high.   

Figure 1: The text in these illustrative items is very hard to read and should be increases in size. Please also add the full names of (at least) the predictors to allow the reader to understand the figure independent from the manuscript text. I also highly recommend that you add similar Figures for the number of endemic species and species turnover as this is not provided though Fig 2 or 3 (where you present predictions not the raw data.

#We apologize for the small letters in the previous version of figures. We enlarged it, and added full names of all variables (lines 281-288). Regarding raw data for endemic, afrotropical (etc.) species, these are in the left columns of the two figures. It was not clear enough in the previous version, so we clarified this at lines 335, and 377.

Figures 2 and 3: The tick marks of the scale are too small.

#The tick marks were ridiculously small indeed. As this was graphically near impossible to enlarge them for so many small maps, and they were redundant anyway, we erased them, and replaced them by an explanation in figure legends, see line 337-8.

Supporting information: Note that temperature variables need to be divided by 10, which should not influence your results but needs to be adjusted in the table [e.g. 166 °C is not correct] and may influence your interpretations whenever you refer to exact values and coefficients (e.g. slopes an intercepts in model results). Please check throughout the manuscript and adapt.

#We are aware that the temperature units for temperature variables are decimals of °C, but made mistake in description of the units in the Appendix. This has changed now. Regarding coefficients etc., as we used the composite variables Clim1–Clim4 (see also the newly added Supplementary figure S1), this is not an issue here. 

Round 2

Reviewer 3 Report

Dear authors,

thank you for addressing my comments and for clarifying the potential issues raised. To me to paper is not suitable for publication. Congratulations!

One last thing: please specify in the table caption that D2 is the explained deviance (coefficient of dermination, goodness of fit measure).